# Thrombin Preconditioning Enhances Therapeutic Efficacy of Human Wharton’s Jelly–Derived Mesenchymal Stem Cells in Severe Neonatal Hypoxic Ischemic Encephalopathy

**DOI:** 10.3390/ijms20102477

**Published:** 2019-05-20

**Authors:** Young Eun Kim, Se In Sung, Yun Sil Chang, So Yoon Ahn, Dong Kyung Sung, Won Soon Park

**Affiliations:** 1Department of Pediatrics, Samsung Medical Center, Sungkyunkwan University School of Medicine, Seoul 06351, Korea; duddms920@skku.edu (Y.E.K.); sein.sung@samsung.com (S.I.S.); yschang@skku.edu (Y.S.C.); soyoon.ahn@samsung.com (S.Y.A.); stem.sung@samsung.com (D.K.S.); 2Stem Cell and Regenerative Medicine Institute, Samsung Medical Center, Seoul 06351, Korea; 3Department of Health Sciences and Technology, SAIHST, Sungkyunkwan University, Seoul 06351, Korea

**Keywords:** mesenchymal stem cell transplantation, hypoxic-ischemic encephalopathy, translational medical research, infant, newborn, diseases

## Abstract

We investigated whether thrombin preconditioning of human Wharton’s jelly–derived mesenchymal stem cells (MSCs) improves paracrine potency and thus the therapeutic efficacy of naïve MSCs against severe hypoxic ischemic encephalopathy (HIE). Thrombin preconditioning significantly enhances the neuroprotective anti-oxidative, anti-apoptotic, and anti-cytotoxic effects of naïve MSCs against oxygen–glucose deprivation (OGD) of cortical neurons in vitro. Severe HIE was induced in vivo using unilateral carotid artery ligation and hypoxia for 2 h and confirmed using brain magnetic resonance imaging (MRI) involving >40% of ipsilateral hemisphere at postnatal day (P) 7 in newborn rats. Delayed intraventricular transplantation of 1 × 10^5^ thrombin preconditioned but not naïve MSCs at 24 h after hypothermia significantly enhanced observed anti-inflammatory, anti-astroglial, and anti-apoptotic effects and the ensuing brain infarction; behavioral tests, such as cylinder rearing and negative geotaxis tests, were conducted at P42. In summary, thrombin preconditioning of human Wharton’s jelly-derived MSCs significantly boosted the neuroprotective effects of naïve MSCs against OGD in vitro by enhancing their anti-oxidative, anti-apoptotic, and anti-cytotoxic effects, and significantly attenuated the severe HIE-induced brain infarction and improved behavioral function tests in vivo by maximizing their paracrine anti-inflammatory, anti-astroglial, and anti-apoptotic effects.

## 1. Introduction

Perinatal hypoxic ischemic-encephalopathy (HIE) still remains a serious disease despite recent advances in perinatal medicine with high mortality and neurologic disabilities in survivors such as cerebral palsy, mental retardation, and learning disabilities [1,2]. Although hypothermia treatment is the only currently available treatment that is known to be effective to improve the outcome of neonatal HIE [3,4] in clinical practice, more than half of HIE infants die or experience significant neurologic complications, especially in severe HIE [5,6]. Therefore, developing new and effective adjuvant treatments besides therapeutic hypothermia to enhance neuroprotective effects and improve the outcome of severe neonatal HIE is an urgent subject requiring significant attention.

Recent studies have reported the neuroprotective effects of mesenchymal stem cells (MSCs) transplantation in neonatal animal models of HIE [7,8], stroke [9], and intraventricular hemorrhage (IVH) [10]. We have previously reported that concurrent or delayed MSCs transplantation with hypothermia treatment synergistically attenuates severe HIE, in contrast to stand alone therapy [11,12]. Moreover, transplantation of autologous umbilical cord blood mononuclear cells to neonates with HIE in addition to hypothermia treatment [12] or allogenic MSCs to neonates with severe IVH [13] in phase I clinical trials has been shown to be safe, feasible, and potentially efficacious. Overall, these findings suggest that cell-based therapies combined with therapeutic hypothermia might act synergistically and could thus be a novel, effective therapy that could improve the outcome of severe neonatal HIE, which is currently intractable.

Recent studies suggest that the pleiotropic anti-inflammatory, anti-fibrotic, anti-oxidative, anti-apoptotic, anti-microbial, and permeability-decreasing protective effects of transplanted MSCs have been known to be mediated predominantly by paracrine action via the secretion of various biologic factors, MSC-derived exosomes, and mitochondria transfer rather than through direct regenerative action [14,15]. There is growing evidence that in vitro MSC preconditioning—including exposure to hypoxia, lipopolysaccharides, growth factors, hormones, and pharmacologic or chemical agents—could optimize their paracrine potency and therapeutic potential [14,16,17]. In our previous study [18], we compared various preconditioning regimens and observed thrombin preconditioning of human umbilical cord blood (UCB)-derived MSCs best accelerated cutaneous wound healing compared with hypoxia, lipopolysaccharide, or H_2_O_2_ preconditioning by boosting exosome biogenesis and enriching their cargo contents. The main reason we tested Wharton’s jelly-derived MSCs instead of UCB-derived MSCs was its logistical advantage with easier harvest, expansion, and large-scale production than UCB-derived MSCs. Another reason was to investigate whether the beneficial effects of thrombin preconditioned UCB-derived MSCs observed in our previous study could be extended to another source of Wharton’s jelly-derived MSCs. In the present study, we thus investigated whether thrombin preconditioning enhances the therapeutic efficacy of human Wharton’s jelly-derived MSCs in vitro for oxygen–glucose deprivation (OGD) and in vivo in severe HIE in newborn rats. 

## 2. Results

### 2.1. Cell Viability, Cytotoxicity, Oxidative Stress, and Cell Death Assay after Oxygen–Glucose Deprivation

In primary cultures of rat cortical neurons in vitro, the oxygen–glucose deprivation (OGD)-induced decrease in cell viability and increases in cytotoxicity, oxidative stress, and cell death—evidenced by cell viability assay and lactate dehydrogenase (LDH), malondialdehyde (MDA), and terminal deoxynucleotidyl transferase dUTP nick end labeling (TUNEL) assays, respectively—were significantly improved with naïve MSCs but not with fibroblast administration. Thrombin-preconditioned MSCs significantly enhanced the neuroprotective beneficial effects of naïve MSCs (Figure 1). The thrombin preconditioning also significantly increased levels of brain-derived neurotrophin factor and vascular endothelial growth factor measured in culture media of thrombin-preconditioned MSCs were compared to naïve MSCs (Appendix A).

### 2.2. Serial Brain Magnetic Resonance Image and Injury Assessment

Figure 2A illustrates a representative serial brain magnetic resonance image (MRI) obtained on postnatal day (*p*) 7 (2 h after HIE) and P42 (35 days after HI) in each experimental group. Although the baseline ipsilateral brain infarct volume on P7 was not significantly different between experimental groups, the intact brain volume in the HIE injury control group (HNC) rats progressively reduced over time, as shown by a follow-up brain MRI performed on P42 (Figure 2B). The reduced intact brain volume observed in HNC on P42 was significantly attenuated in the combined treatment of hypothermia and thrombin-preconditioned MSCs group (HHT), but not in the hypothermia treatment alone group (HHC) or combined hypothermia and naïve MSCs group (HHM).

### 2.3. Brain Cell Death and Reactive Gliosis

A marked increase in the number of TUNEL-positive cells was observed in all HIE groups compared to the normal control group (NNC) rats in the peri-infarct brain area on P10 (Figure 3A,D); also, many presence of neuronal nuclei- and TUNEL- double positive cells were detected in the peri-infarct brain area. HIE-induced increase in cell death was significantly attenuated in HHT but not in HHC or HHM. 

Elevated glial fibrillary acidic protein (GFAP) level, indicative of reactive gliosis, was observed in all HIE groups compared with NNC (Figure 3B,E). The enhanced gliosis observed in HNC was significantly attenuated both in HHM and in HHT, but not in HHC. 

### 2.4. Brain Inflammation

The number of ED-1-positive cells, indicative of activated microglia, significantly increased in all HIE groups compared to NNC in the peri-infarct brain area on P10, and this increased number of ED-1 positive microglial cells observed in HNC was significantly attenuated in HHT but not in HHC or HHM (Figure 3C,F).

The levels of pro-inflammatory cytokines such as interleukin (IL)-1α, IL-1β, IL-6, and tumor necrosis factor (TNF)-α were significantly increased in all HIE groups compared to NNC (Figure 4). Increased levels of IL-1α and IL-6 observed in HNC were significantly attenuated both in HHM and HHT but not in HHC; the increased levels of IL-1β and TNF-α observed in HNC were significantly attenuated only in HHT but not in HHC or HHM. 

### 2.5. Functional Behavior Tests

To evaluate sensorimotor functions of the HIE rats, the cylinder test and Y-maze test were performed at 5 weeks after HIE induction (Figure 5A,C), the negative geotaxis test was performed weekly until 5 weeks after HIE induction (Figure 5B), and the rotarod test was performed for three consecutive days before the end of the 5-week experiment (on postnatal days 40, 41, and 42) (Figure 5D).

In the cylinder test, limb-use asymmetry was significant in all the HIE groups compared to NNC, and limb-use asymmetry observed in HNC was significantly improved only in HHT but not in HHC or HHM (Figure 5A). In the negative geotaxis test, NNC presented a short duration for rotating, indicating a quick response time and intact sensorimotor function, and the HNC displayed a significantly longer duration, indicative of impaired function, than NNC (Figure 5B). The prolonged negative geotaxis test observed in HNC at 5th week after HIE induction was significantly improved both in HHM and HHT but not in HHC. All HIE groups showed significantly impaired Y-maze test (Figure 5C) and rotarod test (Figure 5D) compared to NNC, and these impaired Y-maze and rotarod tests observed in HCN were not significantly improved in HHC, HHM, or HHT. 

## 3. Discussion

Perinatal hypoxic ischemic encephalopathy (HIE) is a serious disorder that exhibits high mortality and neurological morbidities in survivors despite hypothermia treatment, especially in the severe type of HIE [5,6]. Therefore, developing an appropriate animal model to simulate clinically severe HIE is an essential first step in determining its pathophysiological mechanisms and testing the therapeutic efficacy of any novel treatments. In this study, we used the well-established Rice–Vannucci model of HIE, comprising unilateral carotid artery ligation followed by exposure to a hypoxic environment in postnatal day (P) 7 rats [19]. Although the Rice–Vannucci model of HIE has a non-clinical distribution of injury, somewhat between the pattern seen with global asphyxia and that of a true stroke [20,21,22], we selected this animal model since large amounts of neuropathologic, biochemical, and long-term functional outcome data continue to accrue from this relatively inexpensive and easily mastered model of HIE, thereby making it most suitable for testing the therapeutic efficacy of MSC transplantation against neonatal HIE. Currently, hypothermia is the only clinically available treatment, and known to be effective against neonatal HIE. However, it is not quite effective, especially in severe neonatal brain injury. Since a lot of previous preclinical data demonstrated neuroprotection from hypothermia in the same newborn rat pup model of HIE [21,23,24,25,26], our current data showing no protection with hypothermia might not be attributable to the animal model alone but rather to the selection of homogenous population of severe neonatal HIE with brain MRI. Therefore, it would be more clinically meaningful to test the therapeutic efficacy of MSC transplantation in the highest risk population with HIE. Although large animals, such as piglets, would be a better selection to mimic neonatal hypoxia-ischemia, the animals are very expensive; hence, only a few of them could be induced for HIE at a time, and few neuropathologic data would be available for comparison, and long-term follow up neurobehavioral testing data would be absent. Considering these advantages and limitations, we considered large animals and the newborn rodent models to not be mutually exclusive but rather complimentary to each other. However, one significant drawback of this model is the wide variability in the severity of HIE injury and the ensuing brain infarct, which limits direct comparison between the experiments [22]. To overcome this drawback, we randomly allocated rat pups into four experimental groups only after confirming the induction of severe HIE involving more than 40% of the ipsilateral hemisphere volume, as confirmed by brain diffusion weighted magnetic resonance imaging (MRI), conducted 2 h after modeling. Since persistent induction of severe HIE, progress of brain infarct, and long-term neurologic impairments have been observed in both present and previous studies [11,12], our results, which showed the progress of severe HIE to brain infarct via in vivo brain MRI, histologic abnormalities, and impaired behavioral function despite hypothermia treatment, indicate that our newborn rat pup model is appropriate for testing the therapeutic efficacy of potentially novel treatments besides hypothermia treatment.

MSCs have exhibited remarkable therapeutic effects in numerous preclinical newborn brain disease models, including HIE [11,12], bacterial meningitis [27], and intraventricular hemorrhage [13]. However, successful clinical translation of MSCs is hampered by their heterogeneity, which leads to high variability in their therapeutic efficacy, depending on their sources [28]. Furthermore, potent MSC therapeutics require MSCs with maximum therapeutic efficacy and regenerative capacity. There is growing evidence that in vitro MSC preconditioning could optimize their paracrine potency and hence their therapeutic potential [14,16,17]. In our previous study [18], thrombin preconditioning of human umbilical cord blood (UCB)-derived MSCs exhibited optimal therapeutic efficacy compared to other preconditioning regimens, such as hypoxia, lipopolysaccharides, or H_2_O_2_, in promoting proangiogenic activity in vitro and enhancing cutaneous wound healing in vivo. In the present study, thrombin preconditioning significantly enhanced the neuroprotective anti-oxidative, anti-apoptotic, and anti-cytotoxic effects of naïve MSCs against oxygen–glucose deprivation in vitro. Moreover, only thrombin preconditioned MSCs—but not naïve MSCs—significantly attenuated the severe HIE-induced brain infarction. Overall, these findings suggest that thrombin preconditioning could maximize the therapeutic efficacy of naïve MSCs. Moreover, thrombin preconditioning neither was cytotoxic nor altered the characteristics of MSCs, including their surface marker profiles and in vitro adipogenesis and osteogenesis; however, it boosted the biogenesis of MSCs-derived exosomes production and enriched their cargo contents via largely protease activated receptor (PAR)-1-mediated and partly PAR-3-mediated Rab5, early endosomal antigen-1, extracellular signal regulated kinase 1/2, and AKT signaling pathways activation [29]. Since human thrombin is clinically available, based on the favorable experimental results of our previous [18] and present studies, we plan to embark on a phase I clinical trial for moderate-to-severe neonatal HIE treatment using thrombin-preconditioned MSCs.

In our previous studies [9,13,14,18,27,30], we had observed the existence of a cross-talk between transplanted MSCs and injured host tissue site. Although the same MSCs were transplanted, the paracrine factors secreted by MSCs mediating their anti-inflammatory; anti-apoptotic; anti-oxidative; and, sometimes, anti-bacterial effects were quite variable in strict response to their local micro-environmental cues. Therefore, we had only tested hypoxia, H_2_O_2_, lipopolysaccharide, and thrombin as preconditioning regimens to simulate the clinical conditions of HIE, bronchopulmonary dysplasia, sepsis, and IVH, respectively, in our previous study [18].

Although numerous mechanisms may be involved, our data showing significantly enhanced anti-oxidative, anti-apoptotic, and anti-astroglial, and anti-inflammatory effects of naïve MSCs after thrombin preconditioning suggest that the mechanism by which preconditioning enhances the therapeutic efficacy of transplanted MSCs seems to be mediated primarily by stimulating the secretion of growth factors, including vascular endothelial growth factor and hepatocyte growth factor, cytokines, and other proteins, as well as by releasing exosomes from MSCs to ensure their maximal paracrine potency [14,16,17,18]. Additionally, in current study, we have additionally observed significantly increased concentrations of trophic factors such as brain-derived neurotrophin factor (BDNF) and vascular endothelial growth factor (VEGF), after thrombin preconditioning in the culture media of human Wharton’s jelly-derived MSCs. The next step of our future study would be to elucidate the precise neuroprotective mechanism of these trophic factors underlying the thrombin preconditioning of MSCs.

In contrast to our previous studies showing anti-inflammatory and anti-apoptotic effects, and the best resultant attenuation of severe HIE-induced brain infarction using combined hypothermia and concurrent or delayed intraventricular transplantation of 1 × 10^5^ human UCB–derived MSCs [11,12], our current data of delayed intraventricular transplantation of 1 × 10^5^ human Wharton’s jelly-derived naïve MSCs combined with hypothermia were not quite as effective in inducing anti-inflammatory and anti-apoptotic effects, or for attenuating the ensuing brain infarction following severe HIE. These findings suggest that the source of MSCs might impact their paracrine potency and thus the therapeutic efficacy of stem cell therapies [28]. Nonetheless, our data also suggest that thrombin preconditioning of MSCs could maximize their paracrine potency and therapeutic efficacy regardless of their origin [18].

Besides improving histologic abnormalities and attenuating brain infarction volume, improving behavioral function is crucial for the clinical translation of delayed transplantation of thrombin preconditioned MSCs combined with hypothermia for treating severe HIE. In the present study, negative geotaxis test at P42 was significantly improved both in naïve and in thrombin-preconditioned MSC transplantation combined with hypothermia treatment. Along with the significantly attenuated brain infarct volume, severe HIE-induced impairment in the cylinder rearing test was significantly improved only in thrombin preconditioned MSC transplantation combined with hypothermia treatment at P42. In concordance with our data, we previously observed a positive correlation between the functional improvements of this asymmetry and the degree of tissue preservation in the injured hemisphere [31]. Furthermore, the improved behavioral function test results observed at P42 imply that the neuroprotective effects of single transplantation of thrombin preconditioned MSCs for severe HIE could persist into adolescence when treating humans [32]. Our data, which show no significant improvements in the Y-maze and rotarod tests despite significant attenuation of the brain infarction volume with thrombin preconditioned MSC transplantation, suggest that, besides intact brain volume, other factors such as improved myelination or the involvement of critical areas, such as the hippocampus, might also be involved in improved sensorimotor function. The data showing more pronounced positive effects in motor functions, while cognitive function in Y-maze is barely improved, suggest that besides improved brain infarct volume, improvements in critical areas such as hippocampus and frontal cortex would be important for improved spatial learning and cognitive flexibility [33]. Further studies should be conducted to clarify this.

For the fate of transplanted MSCs, the number of transplanted MSCs drastically reduced by 72 h after intranasal transplantation [34]; <1% was detected at 18 days after intracranial administration [35], and virtually no MSCs were detected at 70 days after intratracheal administration [36]. Since the transplanted MSCs are known not to engraft in the brain, and exert therapeutic function through a brief ‘hit and run’ mechanism [37], there remains no concern about long-term adverse effects including tumor formation. Our data of sustained long-term neuroprotection without any long-term adverse effect warrant the translation of MSC transplantation into clinical studies for the treatment of neonatal HIE.

In summary, thrombin preconditioning of human Wharton’s jelly-derived MSCs significantly boosted the neuroprotective effects of naïve MSCs against oxygen–glucose deprivation in vitro by enhancing their anti-oxidative, anti-apoptotic, and anti-cytotoxic effects and significantly attenuated severe HIE-induced brain infarction and improved behavioral function tests by maximizing their paracrine anti-inflammatory, anti-astroglial, and anti-apoptotic effects.

## 4. Materials and Method

### 4.1. Cell Preparation

The study was approved by the Institutional Review Board of Samsung Medical Center. The Wharton’s jelly-derived MSCs were kindly provided by Professor Chang JW in charge of good manufacturing practice facility at Samsung Stem Cells and Regenerative Medicine Institute. After informed consent was obtained from pregnant mothers, human Warton’s jelly-derived MSCs were isolated and expanded, as described previously [38]. Human Warton’s jelly-derived MSCs from a single donor at passage 5–6 were used in this study. Stemness of MSCs was confirmed using in vitro differentiation assays into osteogenesis, adipogenesis, and chondrogenesis and flow-cytometric analysis for cell surface markers (CD73, CD90, CD105, CD166, CD14, CD11b, HLA-DR (MHCII), CD34, CD45, and CD19), as described previously [39]. After getting 90% confluence, MSCs were preconditioned with thrombin (2 U/mL; Sigma–Aldrich, Steinheim, Germany) in culture medium (α-MEM; Gibco, Life Technologies, Carlsbad, CA, USA) for 3 h. Control naïve MSCs were prepared in the same manner except for the thrombin treatment.

### 4.2. In Vitro Model of Oxygen–Glucose Deprivation

Cerebral cortical neurons were isolated at embryonic day 18–19 in rats, as described previously [40]. At day 10 of primary culture, oxygen–glucose deprivation (OGD) was performed to induce cortical neuronal cell death [40]. Briefly, the cortical neurons were bubbled with 95% N2/5% CO_2_ in glucose free media. The cortical neurons were transferred to an anaerobic chamber (Galaxy 48R incubator; Eppendorf/Galaxy Corporation, Enfield, CT, USA) containing 1% of O_2_ and 5% of CO_2_ humidified at 37 °C, which was then maintained at a constant pressure of 1500 Pa for 90 min. OGD was terminated by replacing the media with Neurobasal culture media containing B27 supplement (Gibco, Grand Island, NY, USA) without antioxidants. Following media replacement, they were returned to the normoxic incubator. Control cultures in a solution identical to the OGD solution but containing glucose (33 mmol/L; control solution) were kept in the normoxic incubator for the same time period as the OGD experiment, and the incubation solution was replaced with reperfusion buffer. Cultures were then returned to the normoxic incubator. To explore the neuroprotective effects of MSCs, 2 U/mL of thrombin induced MSCs, 2 × 10^4^ of MSCs, and MRC-5 fibroblast (Seoul, Korea; KCLB No. 10171) as a control cell line were added in to the upper chamber of an insert with a 1-μm pore (Falcon; Corning Inc., Corning, NY, USA) following OGD and reoxygenation.

### 4.3. In Vitro Cell Viability and Cytotoxicity Assays

Twenty four hours after the cortical neurons were incubated with MSCs or fibroblasts, the cell counting kit (CCK)-8 (Dojindo, Kumamoto, Japan) assay was conducted according to the manufacturer’s instructions to determine the relative cell proliferation rate (%) of the cortical neurons. Cytotoxicity was determined by lactate dehydrogenase (LDH) releases, according to the supplier’s instructions (Roche, Mannheim, Germany). Duplicate measurements were performed for each sample.

### 4.4. In Vitro Oxidative Stress Assay

The level of malondialdehyde (MDA), as a marker of oxidative stress, was measured in cell lysates in duplicate, using the Oxiselect TBARS assay kit containing thiobarbituric acid-reactive substances (Cell Biolabs, San Diego, CA, USA), according to the manufacturer’s instructions.

### 4.5. In Vivo Model of Hypoxic Ischemic-Encephalopathy

All animal protocols were reviewed by the Institutional Animal Care and Use Committee (IACUC), and the animals were housed in our AAALAC-approved facility (Samsung medical center). Male sprague-Dawley rats (Orient Co., Seoul, Korea) at postnatal day (P) 7 were raised with their dam rats in the standard cage, except during the hypothermia period. Dam rats had free access to water and chow in an alternating 12-h light/dark cycle in a constant humidity and temperature environment. We assessed and monitored the condition of rat pups regularly, four times per day during the animal studies. As we previously described [12], HIE was induced by ligation of unilateral (right side) carotid artery and exposure to 8% oxygen for 2 h. The rat pups were randomly allocated into five experimental groups in a blind manner as follows: normal (sham) with NNC; HIE with HNC; HIE with HHC; HIE with HHM, and HIE with HHT. Induction of severe HIE involving more than 40% of the ipsilateral hemisphere volume was confirmed by magnetic resonance imaging (MRI) of brain after 2 h of modeling, and then 24 h of hypothermia intervention followed in a hyperthermia chamber set at 32 °C within 3 h after HIE induction. As a control, normothermia chamber was set at 36 °C for the same period. Throughout the 24 h of temperature intervention, pups were fed five times with 0.5 mL of milk formula using a 22 gauge feeding needle. During the temperature intervention period, rectal temperatures of rat pups were monitored (Appendix A). Brain tissues were obtained for histological and biochemical analyses at P11 (3 days after HIE induction). In long-term follow-up study, MRI was performed at P42 (5 weeks after HIE induction), and functional behavioral tests were performed from the first week to the fifth week after HIE induction, depending on the timing of the functional test. Figure 6 displays the experimental schedule and groups in the present study.

### 4.6. Transplantation of Mesenchymal Stem Cells

The dose of transplanted MSCs (1 × 10^5^) was chosen based on our previous findings [11,12], in which MSCs showed significant protective effects on HIE in newborn rats. Briefly, 1 × 10^5^ MSCs in 10 μL saline were administered into the ipsilateral right lateral ventricle using a stereotactic method (Digital Stereotaxic Instrument with fine drive, MyNeurolab, St. Louis, MO, USA; coordinates, × = +0.5, y = +1.2, z = −2.7 mm relative to bregma), as described previously [11], following 24 h of hypothermia treatment after HIE. An equivalent volume (10 μL) of saline was administered by the same administration protocol. 

### 4.7. Brain Intact Volume Assessment

Brain MRI was used to confirm initial brain injury after HIE induction on postnatal day 7 and to monitor changes in the brain injury 5 weeks after HIE induction. MRI was performed using a 7.0-Tesla MRI system (Bruker-Biospin, 8117 Fällanden, Switzerland), as described previously [11,13]. Initial lesions were identified as hyperintense areas in diffusion-weighted imaging performed after 2 h of HIE induction, and final lesions were identified as hyperintense areas in T2-weighted imaging conducted 5 weeks after HIE induction. The intact ipsilateral-to-whole-contralateral hemispheric volume ratio was calculated as a measure of brain injury, as previously reported [11]. The researcher performing this procedure was blinded to the treatment group.

### 4.8. Immunohistochemistry

Reactive gliosis and microglia activation levels were histologically evaluated with immunohistochemistry staining for glial fibrillary acidic protein (GFAP) (Z0334, Dako, Glostrup, Denmark) as an astrocytic glial marker and ED-1 (Ab31630, Abcam, Cambridge, United Kingdom) as a reactive microglial marker. The optical density of GFAP and number of ED-1-positive cells were measured in a blinded manner on three non-overlapping fields in three coronal sectioned brains (+0.95 mm to −0.11 mm/bregma) in the peri-infarct area of each brain. 

### 4.9. TUNEL Assay

Brain cell death was evaluated via a terminal transferase-mediated biotin dUTP nick end labeling (TUNEL) assay (kit G3250, Promega, Madison, WI, USA), according to the manufacturer’s protocol. TUNEL positive cells were counted in a blinded manner on three non-overlapping fields in three coronal sectioned brains (+0.95 mm to −0.11 mm/bregma) in the peri-infarct area of each brain. 

### 4.10. Enzyme Linked Immunosorbent Assay

Levels of brain-derived neurotrophin factor and vascular endothelial growth factor were measured in culture media of human Wharton’s jelly-derived MSCs using a Procataplex Multiplex ELISA Kit according to the manufacturer’s protocol (eBioscience, Wien, Austria). Levels of brain inflammatory cytokines, such as of interleukin (IL)-1α, IL-1β, IL-6, and tumor necrosis factor (TNF)-α, were measured in peri-infarct area of the brain tissues using a MILLIPLEX MAP ELISA Kit according to the manufacturer’s protocol (EMD Millipore, Billerica, MA, USA).

### 4.11. Functional Behavioral Tests

The Y-maze test was performed 5 weeks after HIE induction (at P42) to assess hippocampal-dependent short-term memory function [41]. The Y-maze consists of a 120° angle of three horizontal arms. After 10 min of adoption for rats maze arms, arm alterations were recorded over 10 min. The rest time between the acclimatization trial and testing trial was about 2 h. Spontaneous alteration was defined as entries into all three arms consecutively. Between the tests, each arm was thoroughly cleaned. Spontaneous alteration was calculated using the following formula: spontaneous alternation = [(number of alternations)/(total entries-2)]. The number of arm entries serves as an indicator of locomotor activity.

The cylinder test was performed at P39 to assess forelimb movement in a transparent cylinder (25 cm diameter and 40 cm height), as previously described [9]. Each animal was video-recorded for 10 min during each session. All measurements were recorded twice. In the test, the first limb to contact the wall was scored as an independent wall placement for that limb. If an animal placed both paws on the wall, both limbs were scored. Limb-use asymmetry was calculated using the following formula: number of left forelimb contacts/number of both contacts. 

The negative geotaxis test, which is based on the innate reflex rotation to face uphill when placed head down on an inclined wooden platform [42], was performed at P14, P21, P28, P35, and P42. Pups were gently held for 3–5 s in a head-downward position on a slanted slope, and the time required for the pups to rotate 180° to face uphill after release was recorded. The pups were observed for up to 60 s, and if the pup could not complete this test in 60 s or fell down from the slope more than three times, the score was recorded as 60 s. All behavioral function tests were performed in a blinded manner.

### 4.12. Statistical Analysis

Data are expressed as mean ± standard deviation (SD). All data had significant normal distribution (Shapiro–Wilk normality test, *p* > 0.05). For continuous variables, one-way analysis of variance (ANOVA) and Tukey’s post hoc test was performed to determine statistical significance between groups. *p*-values under 0.05 were considered statistically significant. All data were analyzed using SPSS version 18.0 (IBM, Chicago, IL, USA).

## Figures and Tables

**Figure 1 ijms-20-02477-f001:**
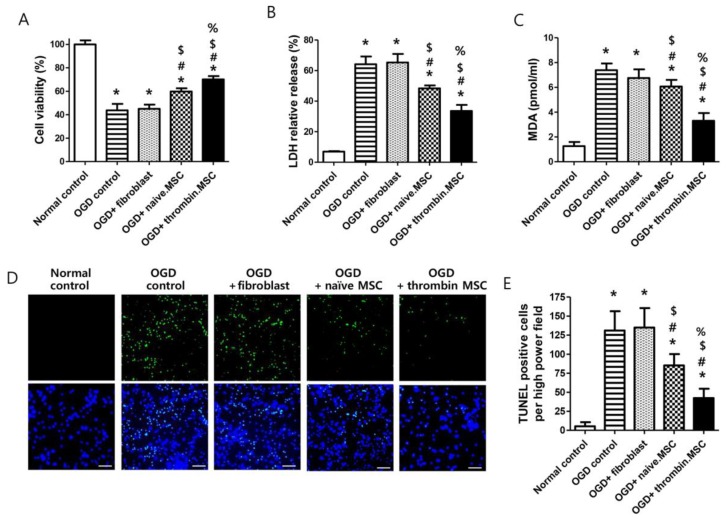
Effect of thrombin priming on neuroprotective efficacy of mesenchymal stem cells (MSCs) in vitro oxygen–glucose deprivation (OGD)-induced primary cultured rat cortical neurons. (**A**) Cell viability, expressed as relative proliferation rate (%) to normal control group; (**B**) cytotoxicity, expressed as relative lactate dehydrogenase (LDH) release (%) to positive control (100% fully killed cells); (**C**) malondialdehyde (MDA) level; and (**E**) the number of terminal deoxynucleotidyl transferase dUTP nick end labeling (TUNEL)-positive cells captured using fluorescent microscopy (**D**) green, original magnification; ×400, scale bars; 20 μm) evaluated in the rat cortical neurons 24 h after co-culture with fibroblasts, naïve MSCs, or thrombin-primed MSCs (*n* = 6/group). Data are expressed mean ± SD. * *p* < 0.05 vs. normal control. # *p* < 0.05 vs. OGD control. $ *p* < 0.05 vs. OGD + fibroblasts. % *p* < 0.05 vs. OGD+naïve MSCs.

**Figure 2 ijms-20-02477-f002:**
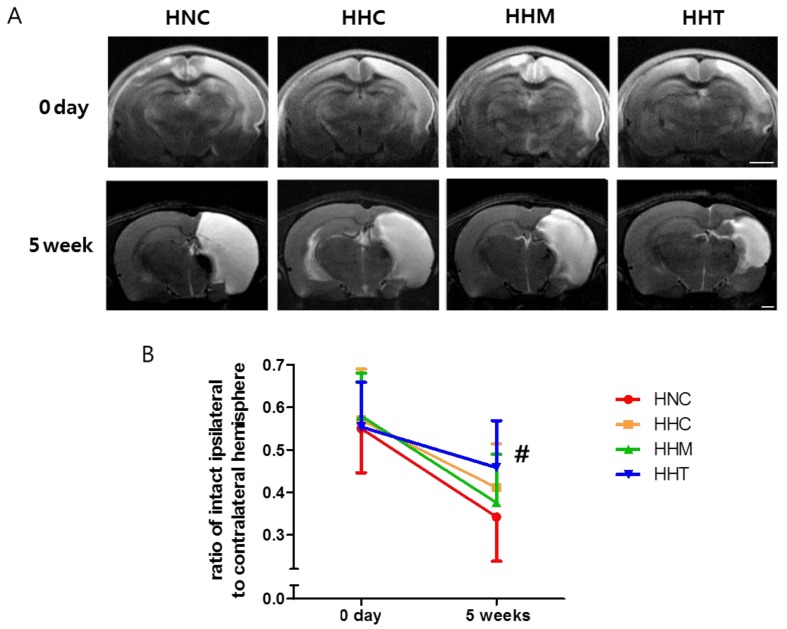
Effect of thrombin-primed MSCs on brain infarction 5 weeks after hypoxic ischemic encephalopathy (HIE). (**A**) Representative brain magnetic resonance images from treatment groups at 0 days (diffusion-weighted image, upper penal) and at 5 weeks (T2-weighted image, lower penal) after HIE induction. Scale bar = 1 mm (**B**) Volume ratio of ipsilateral intact area to the contralateral whole brain area measured by magnetic resonance images (*n* = 17, 21, 16, and 14 in the HIE injury control group (HNC), hypothermia treatment alone group (HHC), combined hypothermia and naïve MSCs group (HHM), and combined hypothermia and thrombin preconditioned MSCs group, respectively). Data are expressed as mean ± SD. # *p* < 0.05 vs. HNC.

**Figure 3 ijms-20-02477-f003:**
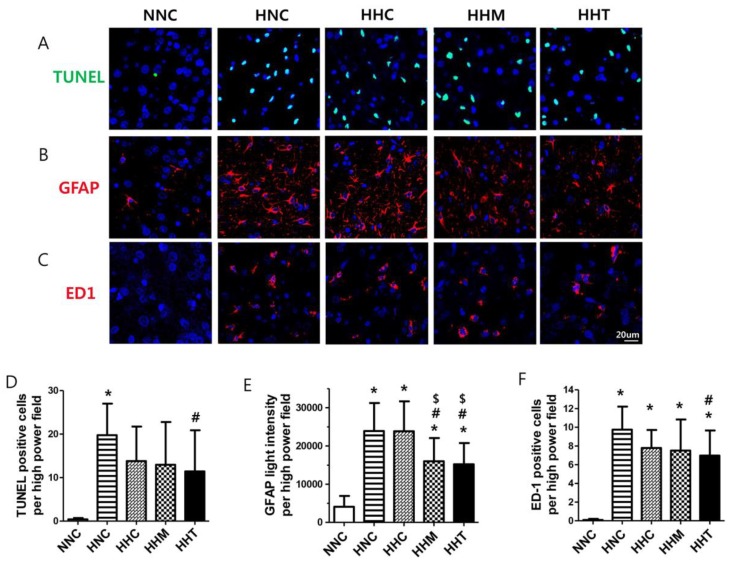
Effects of thrombin-primed MSCs on brain cell death, reactive gliosis, and activated microglia 3 days after HIE. Representative fluorescence micrographs of the penumbra area with staining for (**A**) TUNEL (green, upper panel), (**B**) glial fibrillary acidic protein (GFAP) (red, mid panel), and (**C**) ED-1 (red, lower panel). DNA was counter-stained with 4′,6-diamidino-2-phenylindole (blue) (original magnification; ×400, scale bars; 20 μm). (**D**) Average number of TUNEL-positive cells, (**E**) average intensity of GFAP, and (**F**) average number of ED-1 positive cells, respectively, in the penumbra area (*n* = 9, 28, 9, 28, and 26 in NNC, HNC, HHC, HHM, and HHT, respectively). Data are expressed as mean ± SD. * *p* < 0.05 vs. NNC, # *p* < 0.05 vs. HNC, $ *p* < 0.05 vs. HHC.

**Figure 4 ijms-20-02477-f004:**
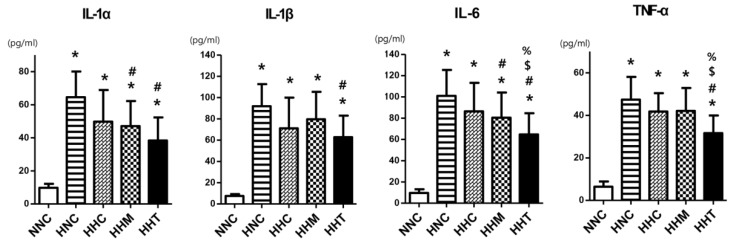
Effects of thrombin-primed MSCs on expression levels of brain inflammatory cytokines 3 days after HIE. Levels of interleukin (IL)-1α, IL-1β, IL-6, and tumor necrosis factor (TNF)-α measured in the penumbra area of brain tissues (*n* = 9, 28, 9, 28, and 26 in normal control group (NNC), HNC, HHC, HHM, and HHT, respectively). Data are expressed as mean ± SD. * *p* < 0.05 vs. NNC; # *p* < 0.05 vs. HNC; $ *p* < 0.05 vs. HHC; % *p* < 0.05 vs. HHM.

**Figure 5 ijms-20-02477-f005:**
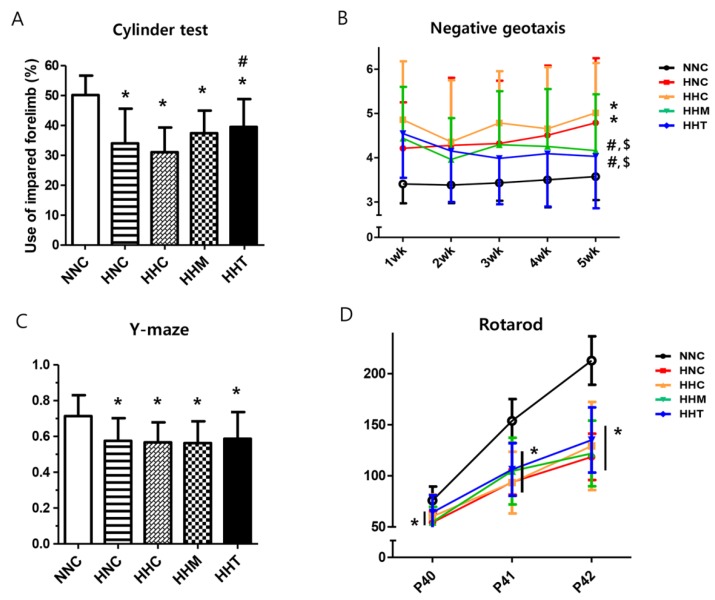
Effects of thrombin-primed MSCs on functional performance capability. Functional performance was assessed using the (**A**) cylinder test 5 weeks after HIE; (**B**) negative geotaxis test 1, 2, 3, 4, and 5 weeks after HIE; (**C**) Y-maze test 5 weeks after HIE; and (**D**) rotarod test for 3 consecutive days 5 weeks after HIE (postnatal days 40–42) (*n* = 10, 17, 21, 16, and 14 in NNC, HNC, HHC, HHM, and HHT, respectively). Data are expressed as mean ± SD. * *p* < 0.05 vs. NNC, # *p* < 0.05 vs. HNC, $, *p* < 0.05 vs. HHC.

**Figure 6 ijms-20-02477-f006:**
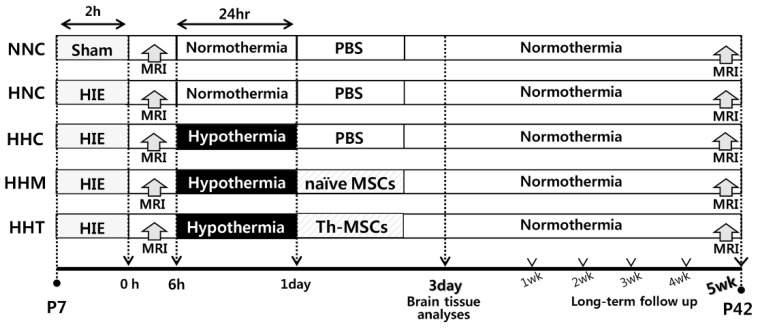
In vivo experimental protocol.

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
