# Peer review of "Thrombin Preconditioning Enhances Therapeutic Efficacy of Human Wharton’s Jelly–Derived Mesenchymal Stem Cells in Severe Neonatal Hypoxic Ischemic Encephalopathy"

_ijms, 2019, doi:10.3390/ijms20102477_

Reviewer 1 Report

This manuscript investigated the effects of thrombin preconditioning human mesenchymal stem cells (MSCs) on the primary culture neuronal cells and severe neonatal hypoxic ischemic encephalopathy (HIE) mouse model. This manuscript is logical and structured.

Strengths of the manuscript are :

1. As indicated by the authors, perinatal HIE is a serious disorder that exhibits high mortality and neurologic morbidities in survivors. Therefore, the manuscript addresses an important clinical problem.

2. The authors show exciting evidence that thrombin may be a novel method to improve MSC function and enhance MSC treatment for HIE

3. The study includes appropriate measures to assess HIE

Nevertheless, there are some aspects that need to be addressed to confirm their hypothesis.

1. As shown, recovery of hypoxic ischemic encephalopathy lesions is attributed to the thrombin preconditioning MSC derived trophic factors. However, there is no attempt to uncover these specific neurotrophic factors.

2. The authors demonstrate that thrombin preconditioning of MSCs significantly boosted the neuroprotective effects of naïve MSCs against OGD in vitro. Therefore the authors should show the TUNEL positive cells are neuronal cells in figure 4. In addition, it would be better to show the expression of pro-apoptotic and anti-apoptotic proteins in neuronal cells and brain tissues

Author Response

 [Reviewer #1]

●1-1) The reviewer stated, “As shown, recovery of hypoxic ischemic encephalopathy lesions is attributed to the thrombin preconditioning MSC derived trophic factors. However, there is no attempt to uncover these specific neurotrophic factors.”

We thank the reviewer for the kind comments on our manuscript. In keeping with the reviewer’s recommendation, we have additionally observed significantly increased concentrations of trophic factors such as brain-derived neurotrophin factor (BDNF) and vascular endothelial growth factor (VEGF) after thrombin preconditioning in the culture media of human Wharton’s jelly-derived MSCs. The next step of our future study would be to elucidate the precise neuroprotective mechanism of these trophic factors underlying the thrombin preconditioning of MSCs. These views were incorporated into the Result, Discussion and Material and method sections (line 80-82, 247-252, 395-397) and Supplementary figure S1 of the revised manuscript. Also, please refer to the response to the editor’s comments.

●1-2) The reviewer stated, “The authors demonstrate that thrombin preconditioning of MSCs significantly boosted the neuroprotective effects of naïve MSCs against OGD in vitro. Therefore the authors should show the TUNEL positive cells are neuronal cells in figure 4. In addition, it would be better to show the expression of pro-apoptotic and anti-apoptotic proteins in neuronal cells and brain tissues.”

We appreciate the reviewer for the insightful comments. Although we could not afford to further evaluate the expression of pro-apoptotic and anti-apoptotic proteins in neuronal cells and brain tissues in this study, we have shown that TUNEL-positive cells in the peri-infarct brain area in Fig. 4 are mostly neuronal cells by demonstrating the co-localization of neuron specific marker NeuN and TUNEL staining in an additional experiment conducted according to the reviewer’s recommendation. This view were incorporated into the Result section (line 115-117) and Supplementary figure S3 of the revised manuscript.

Reviewer 2 Report

 This is a study in which human MSCs are employed to study a rat model of perinatal hypoxic ischemic-encephalopathy for a palliative effect for these cells.  The study is well designed and well written.  However, there are a few minor details that require attention by the authors.

First, the figures should be numbered consecutively.  Figure 2, the last figure in the Methods section, should be numbered accordingly.

A large number of abbreviations are used in the text.  It would be helpful for the reader if these were to be defined up front.  An abbreviation section appears at the end of the Discussion, but the listed abbreviations appear to be irrelevant.  This need to be fixed.

Author Response

[Reviewer #2]

● The reviewer stated, “This is a study in which human MSCs are employed to study a rat model of perinatal hypoxic ischemic-encephalopathy for a palliative effect for these cells.  The study is well designed and well written.  However, there are a few minor details that require attention by the authors.”

We thank the reviewer for the kind comments on our manuscript, and we have made appropriate corrections in our revised manuscript according to the reviewer’s recommendation.

●2-1) The reviewer stated, “First, the figures should be numbered consecutively.  Figure 2, the last figure in the Methods section, should be numbered accordingly.”

We have renumbered consecutively including Fig. 2 in our revised manuscript.

●2-2) The reviewer stated, “A large number of abbreviations are used in the text. It would be helpful for the reader if these were to be defined up front. An abbreviation section appears at the end of the Discussion, but the listed abbreviations appear to be irrelevant. This need to be fixed.”

We tried to reduce the abbreviations used in the text. Also, abbreviations in our revised manuscript have been defined in parentheses the first time they appear in the abstract, main text, and in figure caption and used consistently thereafter.

Reviewer 3 Report

The authors studied advantages of thrombin preconditioning of Wharton’s-jelly derived mesenchymal stem cells on neonatal hypoxia ischemia. There are several problems with this study.

1)     While the authors properly selected animals’ age the method of modeling does not reproduce typical features of neonatal hypoxia-ischemia, which is characterized by the global, bilateral distribution of neonatal ischemic changes. In such case both carotid arteries are temporary closed. Instead, the authors performed model of ischemic stroke. While, potentially this model is valid it has very limited clinical relevance due to rarity of strokes in this group of patients.

2)     Hypothermia is an effective treatment of mild and moderate neonatal hypoxia/ischemia, but it failed to show efficacy in stroke, which might be the reason for the lack of efficacy in current study.

3)     Large animals such as piglets would be better selection to mimic neonatal hypoxia-ischemia.

4)     The figure 2 is missing

5)     In the figure 3A the HHM group is nearly like HHT group, while on the figure 3B HHM groups seems to be even worse than HHC group. Overall, it looks like the images are rather not representative of the whole sample.

6)     Variability of results should be presented on graphs as SDs, not SEMs.

7)     Interestingly, the positive effects are more pronounced in motor functions, while cognitive function in Y-maze is barely improved. It should be discussed.

8)     The lack of results or at least discussion of a fate of transplanted cells (at least biodistribution and survival) is a serious limitation. As it is presented currently MSCs are largely a black box while there is continuous progress in precision of cell transplantation. There are new approaches with 3D printing of cerebral ventricles to study cell biodistribution, as well as there is growing number of approaches to report on cell survival, and labeling of MSCs with fluorine is a very interesting approach, which seems to be relatively harmless for cells.

9)     The authors previously studied several approaches to preconditioning of MSCs however, they did not mention/referred to ascorbic acid, which is frequently used to improve therapeutic properties of MSCs.

10)  The authors should use non-parametric statistics for such small animal groups.

Summing up, the use of local brain injury as a model of global, bilateral brain disease is the most important limitation of the study, while ischemic stroke is enormously rare in this aged group.

Author Response

[Reviewer #3]

● The reviewer commented that the authors studied advantages of thrombin preconditioning of Wharton’s-jelly derived mesenchymal stem cells on neonatal hypoxia ischemia. There are several problems with this study.

●3-1) The reviewer stated, “While the authors properly selected animals’ age the method of modeling does not reproduce typical features of neonatal hypoxia-ischemia, which is characterized by the global, bilateral distribution of neonatal ischemic changes. In such case both carotid arteries are temporary closed. Instead, the authors performed model of ischemic stroke. While, potentially this model is valid it has very limited clinical relevance due to rarity of strokes in this group of patients.”

We thank the reviewer for the insightful comments. Bilateral common carotid artery ligation without accompanying hypoxia seems to be a stronger model of white matter damage (New Reference #20-22). Although we agree with the reviewer’s opinion that the Rice-Vannucci model of HIE, comprising unilateral carotid artery ligation followed by exposure to a hypoxic environment in P7 rats has a non-clinical distribution of injury, somewhat between the pattern seen with global asphyxia and that of a true stroke (New Reference #20-22), we have chosen this animal model despite this limitation as large amounts of neuropathologic, biochemical and long-term functional outcomes data continue to accrue from this relatively inexpensive and easily mastered model of HIE, thereby making it most suitable for testing the therapeutic efficacy of MSCs transplantation against neonatal HIE. These views were incorporated into the Discussion section (line 182-187) of the revised manuscript.

●3-2) The reviewer stated, “Hypothermia is an effective treatment of mild and moderate neonatal hypoxia/ischemia, but it failed to show efficacy in stroke, which might be the reason for the lack of efficacy in current study.”

We thank the reviewer for giving us the opportunity to explain more in detail about the HIE model used in this study. Currently, hypothermia is the only clinically available, and known to be effective treatment for neonatal HIE. However, it’s not quite effective especially for severe neonatal brain injury. As a lot of previous preclinical data demonstrated neuroprotection from hypothermia in the same newborn rat pup model of HIE (New Reference #21 and 23-26), our current data showing no protection with hypothermia might not be attributable to the animal model alone, and rather to the selection of homogenous population of severe neonatal HIE with brain MRI. Therefore, it would be more clinically meaningful to test the therapeutic efficacy of MSCs transplantation in the highest risk population of HIE. These views were incorporated into the Discussion section (line 187-194) of the revised manuscript.

●3-3) The reviewer stated, “Large animals such as piglets would be better selection to mimic neonatal hypoxia-ischemia.”

→ We thank the reviewer for the sensible comments. Although we agree with the reviewer’s opinion that large animals such as piglets would be a better selection to mimic neonatal hypoxia-ischemia, the animals are very expensive. So, only a few animals could be induced to HIE at the same time, and few neuropathologic data are available for comparison, and long-term follow up neurobehavioral testing data are absent. Considering these advantages and limitations, we think the large animals and the newborn rodent models are not mutually exclusive, but complimentary to each other. These views were incorporated into the Discussion section (line 194-200) of the revised manuscript.

●3-4) The reviewer stated, “The figure 2 is missing.”

We have renumbered the figures consecutively in the revised manuscript.

●3-5) The reviewer stated, “In the figure 3A the HHM group is nearly like HHT group, while on the figure 3B HHM groups seems to be even worse than HHC group. Overall, it looks like the images are rather not representative of the whole sample.”

We have replaced the images in each group to represent the whole sample of each group in our revised manuscript according to the reviewer’s recommendation.

●3-6) The reviewer stated, “Variability of results should be presented on graphs as SDs, not SEMs.”

We’ve presented the results on the graphs as SD, instead of SEM, in our revised manuscript.

●3-7) The reviewer stated, “Interestingly, the positive effects are more pronounced in motor functions, while cognitive function in Y-maze is barely improved. It should be discussed.”

We thank the reviewer for the helpful comments. Our data showing more pronounced positive effects in motor functions, while cognitive function in Y-maze is barely improved suggest that besides improved brain infarct volume, improvements in critical areas such as hippocampus and frontal cortex would be important for improved spatial learning and cognitive flexibility (New Reference #33). These views were incorporated into the Discussion section (line 279-282) of the revised manuscript.

●3-8) The reviewer stated, “The lack of results or at least discussion of a fate of transplanted cells (at least biodistribution and survival) is a serious limitation. As it is presented currently MSCs are largely a black box while there is continuous progress in precision of cell transplantation. There are new approaches with 3D printing of cerebral ventricles to study cell biodistribution, as well as there is growing number of approaches to report on cell survival, and labeling of MSCs with fluorine is a very interesting approach, which seems to be relatively harmless for cells.”

For the fate of transplanted MSCs, the number of transplanted MSCs drastically reduced by 72 h after intranasal transplantation (New Reference #34),<1% was detected at 18 days after intracranial administration (New Reference #35), and virtually no MSCs were detected at 70 days after intratracheal administration (New Reference #36). Since the transplanted MSCs do not engraft in the brain, and exert therapeutic function through a brief ‘hit and run’ mechanism (New Reference #37), there’s no concern about long-term adverse effects including tumor formation. Our data of sustained long-term neuroprotection without any long-term adverse effects warrant the translation of MSCs transplantation into clinical studies for treatment of neonatal HIE. These views were incorporated into the Discussion section (line 284-291) of the revised manuscript.

●3-9) The reviewer stated, “The authors previously studied several approaches to preconditioning of MSCs however, they did not mention/referred to ascorbic acid, which is frequently used to improve therapeutic properties of MSCs.”

→ We thank the reviewer for the insightful comments. The reasons we have not tested ascorbic acid, frequently used to improve therapeutic properties of MSCs, in our previous study are as follows; In our previous study (Pediatr Res. 2018 Jan;83:214-222), we observed there’s a cross-talk between transplanted MSCs and injured host tissue site. Although the same MSCs were transplanted, the paracrine factors secreted by MSCs mediating their anti-inflammatory, anti-aopototic, anti-oxidative and sometimes anti-bacterial effects were quite variable in strict response to their local micro-environmental cues. So we only tested hypoxia, H2O2, lipopolysaccharide and thrombin as preconditioning regimens just to simulate the clinical conditions of HIE, bronchopulmonary dysplasia, sepsis and intraventricular hemorrhage, respectively and cross-talk between injured host tissue site and transplanted MSCs in our previous study (J Clin Med. 2019 Apr 18;8. pii: E533). These views were incorporated into the Discussion section (line 234-240) of the revised manuscript. Please also refer to our response to the editor’s comments.

●3-10) The reviewer stated, “The authors should use non-parametric statistics for such small animal groups.”

→ In statistical analyses, according to statistical expert advice we used normality test to determine if our all data set is well-modeled by a normal distribution and to compute how likely it is for a random variable underlying the data set to be normally distributed. After ensuring that all data had significant normal distribution (Shapiro-Wilk normality test, p > 0.05), we performed one-way analysis of variance (ANOVA) for continuous variables to determine statistical significance between groups. We have incorporated this into Material and method section in our revised manuscript (line 424-425)

● The reviewer stated, "Summing up, the use of local brain injury as a model of global, bilateral brain disease is the most important limitation of the study, while ischemic stroke is enormously rare in this aged group."

→ Despite their limitations, based on our response to the reviewer’s comments, we assume that the newborn Rice Vannucci model of unilateral carotid artery ligation and global hypoxia used in this study is a useful model for testing therapeutic efficacy of MSCs transplantation besides hypothermia treatment.

Reviewer 4 Report

Overall, this work follows a series of studies by Park WS and colleagues, investigating the neuroprotective effect of mesenchymal stem cells (MSCs) for neonatal hypoxic ischemic encephalopathy (HIE). Previously, they demonstrated that the combination with hypothermia treatment and umbilical cord blood (UCB)-derived MSCs injection effectively improve the prognosis of severe HIE. In addition, they also reported that thrombin preconditioning facilitated extracellular vesicles production from UCB-derived MSCs, which can stimulate angiogenesis in vitro and cutaneous wound healing in vivo. Based on these findings, in this present study, the authors investigated the therapeutic effect of Wharton’s jelly (Wj)-derived MSCs preconditioned with thrombin on severe HIE in newborn rat.

As a result, they clearly demonstrated that the combination with hypothermia and thrombin conditioned but not naïve Wj-derived MSCs transplantation attenuated HIE-induced brain infraction. The manuscript is well written and certainly the idea merits publication for International Journal of Molecular Sciences. However, a major drawback drastically reduce the impact of this work as described below.

Major points

1. It is unclear why they employed Wj- but not UCB-derived MSCs. Their previous study demonstrated the effectiveness of both hypothermia treatment and thrombin conditioning on UCB-derived MSCs. There is no evidence indicating such beneficial effect on Wj-MSCs. Clear explanation is indispensable in Introduction section. According to their brilliant previous studies, the reviewer assumes that thrombin preconditioned UCB-derived MSCs could be better candidate than Wj-derived MSCs for HIE treatment.

2. In the materials and methods section, they stated that Wj-MSCs were isolated based on the previous report (reference #25). In addition, they cited the previous study (reference #26) for the confirmation of the stemness of their Wj-MSCs. However, the reference #25 and #26 may be published from other group (because the authors in this present study were not included in #25 and #26). This is not acceptable. If the cells were gifted from other group, they should state this point. If the authors isolated and expanded human Wj-MSCs by themselves, they have to show the data confirming the stemness or cite a previous study by their group.

Minor points

1. What does the “%” indicate in figure 5? There is no information about “%” in the figure legend.

Author Response

[Reviewer #4]

● The reviewer commented that overall, this work follows a series of studies by Park WS and colleagues, investigating the neuroprotective effect of mesenchymal stem cells (MSCs) for neonatal hypoxic ischemic encephalopathy (HIE). Previously, they demonstrated that the combination with hypothermia treatment and umbilical cord blood (UCB)-derived MSCs injection effectively improve the prognosis of severe HIE. In addition, they also reported that thrombin preconditioning facilitated extracellular vesicles production from UCB-derived MSCs, which can stimulate angiogenesis in vitro and cutaneous wound healing in vivo. Based on these findings, in this present study, the authors investigated the therapeutic effect of Wharton’s jelly (Wj)-derived MSCs preconditioned with thrombin on severe HIE in newborn rat.

As a result, they clearly demonstrated that the combination with hypothermia and thrombin conditioned but not naïve Wj-derived MSCs transplantation attenuated HIE-induced brain infraction. The manuscript is well written and certainly the idea merits publication for International Journal of Molecular Sciences. However, a major drawback drastically reduces the impact of this work as described below.

Major points

●4-1) The reviewer stated, “It is unclear why they employed Wj- but not UCB-derived MSCs. Their previous study demonstrated the effectiveness of both hypothermia treatment and thrombin conditioning on UCB-derived MSCs. There is no evidence indicating such beneficial effect on Wj-MSCs. Clear explanation is indispensable in Introduction section. According to their brilliant previous studies, the reviewer assumes that thrombin preconditioned UCB-derived MSCs could be better candidate than Wj-derived MSCs for HIE treatment.”

We apologize for our lack of explanation for using Wharton’s jelly-derived MSCs instead of our previously used UCB-derived MSCs in this study. The main reason we tested Wharton’s jelly- derived MSCs instead of UCB-derived MSCs was its logistical advantage with easier harvest, expansion and large scale production than UCB-derived MSCs. Another reason was to investigate whether the beneficial effects of thrombin preconditioned UCB-derived MSCs observed in our previous study could be extended to other source of Wharton’s jelly-derived MSCs. These views were incorporated into the Introduction section (line 64-68) of the revised manuscript.

●4-2) The reviewer stated, “In the materials and methods section, they stated that Wj-MSCs were isolated based on the previous report (reference #25). In addition, they cited the previous study (reference #26) for the confirmation of the stemness of their Wj-MSCs. However, the reference #25 and #26 may be published from other group (because the authors in this present study were not included in #25 and #26). This is not acceptable. If the cells were gifted from other group, they should state this point. If the authors isolated and expanded human Wj-MSCs by themselves, they have to show the data confirming the stemness or cite a previous study by their group.

We apologize again our lack of enough explanation about the Wharton’s jelly-derived MSCs used in this study. The Wharton’s jelly-derived MSCs used in this study were kindly provided by Professor Chang JW in charge of good manufacturing practice (GMP) facility at Samsung Stem Cell and Regenerative Medicine Institute, and the same author of references #38 and #39 (previously #25 and #26) explaining about the same MSCs. We have incorporated these views into Material and method section (line 300-302) in our revised manuscript.

Minor points

●4-3) The reviewer stated, “What does the “%” indicate in figure 5? There is no information about “%” in the figure legend.”

“%” in figure indicates the statistical significance (p-value less than 0.05) between HHM and another group. We have incorporated this into the figure legend (line 148) in our revised manuscript.

Round  2

Reviewer 3 Report

The authors correctly responded to my critique

Reviewer 4 Report

The authors have adequately addressed the issues raised in the review.